# On Quadratic Interpolation of Image Cross-Correlation for Subpixel Motion Extraction [note 1]

**DOI:** 10.3390/s22031274

**Published:** 2022-02-08

**Authors:** Bian Xiong, Qinghua Zhang, Vincent Baltazart

**Affiliations:** 1COSYS-SII, Université Gustave Eiffel, IFSTTAR, 44344 Bouguenais, France; bian.xiong@univ-eiffel.fr (B.X.); vincent.baltazart@univ-eiffel.fr (V.B.); 2Inria, 35042 Rennes, France

**Keywords:** subpixel motion extraction, digital image correlation, quadratic surface fitting, subpixel refinement

## Abstract

Digital image correlation techniques are well known for motion extraction from video images. Following a two-stage approach, the pixel-level displacement is first estimated by maximizing the cross-correlation between two images, then the estimation is refined in the vicinity of the cross-correlation peak. Among existing subpixel refinement methods, quadratic surface fitting (QSF) provides good performances in terms of accuracy and computational burden. It estimates subpixel displacement by interpolating cross-correlation values with a quadratic surface. The purpose of this paper is to analytically investigate the QSF method. By means of counterexamples, it is first shown in this paper that, contrary to a widespread intuition, the quadratic surface fitted to the pixel-level cross-correlation values in the neighborhood of the cross-correlation peak does not always have a maximum. The main contribution of this paper then consists in establishing the mathematical conditions ensuring the existence of a maximum of this fitted quadratic surface, based on a rigorous analysis. Algorithm modifications for handling the failure cases of the QSF method are also proposed in this paper, in order to consolidate it for subpixel motion extraction. Experimental results based on two typical types of images are also reported.

## 1. Introduction

Computer vision techniques for motion extraction are widely developed in a huge variety of applications, including motion tracking, motion compensation, image registration [1], remote sensing [2], biomedicine [3], satellite imagery [4] and vibration analysis [5]. Within this scope, techniques of digital image correlation (DIC) are known to provide accurate results with a high computational efficiency, along with good robustness against noises. Various variants of DIC exist in the literature, for example, phase-only correlation (POC) [2], upsampling cross-correlation (UCC) [6], Fourier-based correlation [1] and virtual image correlation [7].

In this paper, a well-known image correlation technique for subpixel motion extraction is analytically investigated. Subpixel accuracy is particularly important for video-based structural health monitoring (SHM) [5,8] and for aerial or satellite imagery [4]. In such applications, the region of interest may be represented by a small number of pixels in the captured images. It is then important to extract both multipixel and subpixel motions information from video images.

A large variety of correlation-based techniques for motion extraction with subpixel accuracy have been compared in [1,2]. The most widespread techniques are performed in a two-step process. At the pixel level, displacement is estimated by maximizing the cross-correlation between two images. To achieve subpixel accuracy, displacement estimation is then refined in the vicinity of the cross-correlation peak. Among such refinement methods, quadratic surface fitting (QSF) provides a good trade-off between accuracy and computational burden, particularly suitable for video-based SHM, as reported in [9]. This method and its variant forms have also been investigated in [1,2,10,11,12,13,14]. Its good performance has been confirmed by our own experiments.

The purpose of this paper is to mathematically analyze the QSF method. By means of counterexamples, it will be shown that, contrary to a widespread intuition, the quadratic surface fitted to the cross-correlation values in the 3×3 pixels neighborhood of the correlation peak does not always have a maximum, despite the fact that the maximum pixel-level cross-correlation value is located at the center of this 3×3 pixels neighborhood. This absence of maximum leads to a failure of the QSF method, which should determine the subpixel displacement by maximizing the fitted quadratic surface interpolating the cross-correlation values in the vicinity of the cross-correlation peak. However, experiences reported by different authors and conducted by ourselves show that usually the QSF method produces satisfactory results. Then it is important to understand the conditions under which this method works correctly. In this paper, the conditions ensuring the existence of a maximum of the fitted quadratic surface will be formally analyzed. Then these conditions will be completed to make sure that the maximum is within the one-pixel vicinity of the pixel-level cross-correlation peak. Solutions will also be proposed to handle the failures cases of the QSF method by constrained optimization ensuring that the estimated subpixel displacement is within one pixel. These modifications apply only when a failure occurs, hence the extra numerical computation cost is insignificant.

This paper is an extended version of the conference paper [15]. It is organized as follows. The considered problem is formulated in Section 2. The QSF method is recalled in Section 3. Examples showing failures of the QSF method are presented in Section 4. The QSF method is then analyzed in Section 5. Handling of the failure cases is proposed in Section 6. Experimental results based on two typical types of images are reported in Section 7. Finally, conclusions are drawn in Section 8.

## 2. Problem Statement

A locally rigid moving object is observed with a camera. It is assumed that the displacement of the observed object is small between successive images, with negligible rotation and negligible motion along the optical axis direction. Correlation processing will focus on a rectangular template, also known as region of interest (ROI), which includes either the whole moving object or some part of the object. The intensity of each pixel at instant *t* is denoted by I(n,m,t) where the integer pair (n,m) indicates the position of the pixel in an image.

Instead of the usual row and column indexes, in this paper the pair (n,m) denotes pixel integer coordinates in a Cartesian system, as illustrated in Figure 1. It will serve both to describe pixel positions in an image and to fit quadratic surfaces in the QSF method. In the second usage, the origin (x,y)=(0,0) corresponds to zero subpixel displacement. This notation choice is more suitable for the surface fitting problem formulated in the Cartesian coordinate system, in agreement with usual mathematical notations.

Motion extraction will be carried out by determining the horizontal and vertical shifts of the selected template between two images captured at instants *t* and t+Δt. The horizontal image shift (in number of pixels) is decomposed into an integer part $\tilde{n}$ and a fractional (subpixel) part *x* with |x|<1, and similarly the vertical image shift is decomposed into an integer part $\tilde{m}$ and a fractional part *y* with |y|<1, so that:
(1a)$$\begin{array}{cc} \; & \mathrm{Total}\;\mathrm{horizontal}\;\mathrm{shift}=\tilde{n}+x \end{array}$$
(1b)$$\begin{array}{cc} \; & \mathrm{Total}\;\;\;\mathrm{vertical}\;\;\;\mathrm{shift}=\;\tilde{m}+y. \end{array}$$

Given two images (frames) captured at time instants *t* and t+Δt, as illustrated in Figure 2, template shifts are usually estimated through a two-step process [9]. The pixel level (integer) shifts $\left(\tilde{n},\tilde{m}\right)$ are first estimated by maximizing the cross-correlation between the two image templates. At the second step, the cross-correlation is somehow interpolated in the vicinity of the cross-correlation peak to estimate the subpixel shifts (x,y).

At the pixel level, let the cross-correlation be denoted by
(2)$$\begin{array}{cc} r(\tilde{n},\tilde{m}) & ≜\sum_{(n,m)\in T}I\left(n,m,t\right)I\left(n-\tilde{n},m-\tilde{m},t+\Delta t\right), \end{array}$$
where T denotes the set of integer pairs (n,m) corresponding to the pixels belonging to the considered template. The dependences on I(n,m,t) and on I(n,m,t+Δt) are omitted in the notation $r(\tilde{n},\tilde{m})$ for a lighter presentation. The search for the cross-correlation peak is formulated as:(3)$$\begin{array}{c} \left({\tilde{n}}^{*},{\tilde{m}}^{*}\right)=\arg\max_{\begin{array}{c} -\tilde{N}\leq \tilde{n}\leq \tilde{N}\\ -\tilde{M}\leq \tilde{m}\leq \tilde{M} \end{array}}r\left(\tilde{n},\tilde{m}\right), \end{array}$$
where $\tilde{N}$ and $\tilde{M}$ are two positive integers specifying the search ranges respectively for horizontal and vertical shifts.

In order to gain subpixel accuracy, at the second step, the cross-correlation $r(\tilde{n},\tilde{m})$ is somehow interpolated for non integer shifts so that the correlation maximization (Equation 3) can be generalized to subpixel shifts.

In the QSF method, this interpolation is made by fitting, in the least squares sense, a second degree polynomial (or, geometrically, a quadratic surface) to the value of $r({\tilde{n}}^{*},{\tilde{m}}^{*})$ and to the 8 neighboring cross-correlation values $r(\tilde{n},\tilde{m})$, namely $r({\tilde{n}}^{*}+n,{\tilde{m}}^{*}+m)$ with n,m∈{−1,0,+1}. Then the maximum of the fitted polynomial yields the estimated subpixel shifts between the two templates [9].

More formally, the integer shifts $\left({\tilde{n}}^{*},{\tilde{m}}^{*}\right)$ being already estimated, let p(x,y) denote the second degree polynomial fitted, in the least squares sense, to $r({\tilde{n}}^{*}+x,{\tilde{m}}^{*}+y)$ for x,y∈{−1,0,+1}. Then, the subpixel shifts are estimated as
(4)$$\begin{array}{c} \left(x^{*},y^{*}\right)=\arg\max_{\left(x,y\right)\in R^{2}}p\left(x,y\right), \end{array}$$
and the estimated total shifts amount to:(5)$$\begin{array}{c} ({\tilde{n}}^{*}+x^{*},{\tilde{m}}^{*}+y^{*}). \end{array}$$

Satisfactory experimental results of this method have been reported by different authors, for example, [1,2,9]. Our own experiments also confirm its good performance compared to other existing methods for subpixel shift estimation. The main purpose of this paper is to consolidate the theoretical basis of this method.

More specifically, the QSF method, as recalled above, assumes implicitly that the second degree polynomial p(x,y) fitted to the nine correlation values $r({\tilde{n}}^{*}+n,{\tilde{m}}^{*}+m)$, with n,m∈{−1,0,+1}, always has a unique global maximum, corresponding to $\left(x^{*},y^{*}\right)$ located in the one-pixel vicinity of (x,y)=(0,0), so that the total shifts as expressed in (Equation 5) do not fall too far from the pixel level optimal shifts $\left({\tilde{n}}^{*},{\tilde{m}}^{*}\right)$. This paper will investigate the following issues.

1.Does the quadratic surface fitted in the QSF method always have a maximum in the one-pixel vicinity of the pixel-level cross-correlation peak?2.If the answer to the first question is no, what are the conditions ensuring that the fitted quadratic surface has a maximum, and moreover, the maximum is located in the one-pixel vicinity of the pixel-level cross-correlation peak?3.What should the algorithm do if the fitted quadratic surface ever has no maximum, or if its maximum is outside the one-pixel vicinity of the pixel-level cross-correlation peak?

## 3. Quadratic Surface Fitting for Subpixel Refinement

The QSF method is recalled in this section before its analysis in the next sections.

Let $\left({\tilde{n}}^{*},{\tilde{m}}^{*}\right)$ be resulting from the pixel level maximization (Equation 3).

The nine integer pairs (n,m), with n,m∈{−1,0,+1}, form a 3×3 grid (The grid G is formed by nine integer pairs organized in three rows and three columns. It is also seen as a set with the integer pairs as elements, so that notations like (−1,1)∈G can be used.):(6)$$\begin{array}{c} G=\left[\begin{array}{ccc} \left(-1,1\right) & \left(0,1\right) & \left(1,1\right)\\ \left(-1,0\right) & \left(0,0\right) & \left(1,0\right)\\ \left(-1,-1\right) & \left(0,-1\right) & \left(1,-1\right) \end{array}\right]. \end{array}$$

Accordingly, the nine cross-correlation values $r({\tilde{n}}^{*}+n,{\tilde{m}}^{*}+m)$ normalized by the maximum cross-correlation value $r({\tilde{n}}^{*},{\tilde{m}}^{*})$ and denoted by:(7)$$\begin{array}{c} \gamma \left(n,m\right)≜\frac{r({\tilde{n}}^{*}+n,{\tilde{m}}^{*}+m)}{r({\tilde{n}}^{*},{\tilde{m}}^{*})} \end{array}$$
form a matrix
(8)$$\begin{array}{c} \Gamma ≜\left[\begin{array}{ccc} \gamma (-1,1) & \gamma (0,1) & \gamma (1,1)\\ \gamma (-1,0) & \gamma (0,0) & \gamma (1,0)\\ \gamma (-1,-1) & \gamma (0,-1) & \gamma (1,-1) \end{array}\right]. \end{array}$$

The central entry of Γ
(9)$$\begin{array}{c} \gamma \left(0,0\right)=\frac{r({\tilde{n}}^{*}+0,{\tilde{m}}^{*}+0)}{r({\tilde{n}}^{*},{\tilde{m}}^{*})}=1 \end{array}$$
is the maximum cross-correlation value $r({\tilde{n}}^{*},{\tilde{m}}^{*})$ normalized by itself, hence this central entry γ(0,0)=1 is the maximum value among all the nine entries of Γ.

The second degree polynomial:(10)$$p_{\theta }\left(x,y\right)=\theta_{1}+\left[\theta_{2}\;\theta_{3}\right]\left[\begin{array}{c} x\\ y \end{array}\right]+\left[\begin{array}{c} x\;y \end{array}\right]\left[\begin{array}{cc} \theta_{4} & \theta_{5}/2\\ \theta_{5}/2 & \theta_{6} \end{array}\right]\left[\begin{array}{c} x\\ y \end{array}\right],$$
with the vector $\theta \in R^{6}$ collecting the scalar coefficients $\theta_{1},\cdots ,\theta_{6}$, is then fitted to the entry values of Γ for (x,y)∈G, by solving the least squares problem:(11)$$\begin{array}{c} \min_{\theta \in R^{6}}\sum_{(n,m)\in G}{\left[\;p_{\theta }\left(n,m\right)-\gamma \left(n,m\right)\right]}^{2}, \end{array}$$
where G is the grid defined in (Equation 6).

Does this fitted second degree polynomial $p_{\theta }\left(x,y\right)$ always have a unique global maximum?

As explained above, the central entry of the matrix Γ, namely γ(0,0)=1, is the maximum value among all the nine entries of Γ. It then seems reasonable to expect that the second degree polynomial (or quadratic surface) $p_{\theta }\left(x,y\right)$ fitted to the nine entries of Γ has a maximum somehow close to the (maximum) central entry of the matrix Γ, which corresponds to the origin (x,y)=(0,0) of the coordinate system characterizing the fitted quadratic surface.

Unfortunately, the fact that the central entry γ(0,0) is the maximum value among the nine entries of Γ does not really ensure that the fitted second degree polynomial $p_{\theta }\left(x,y\right)$ always has a global maximum, as demonstrated by the following counterexamples.

## 4. Counterexamples

Three examples with either synthetic or real-world images are presented below to show that the quadratic surface fitted in the QSF method does not always have a maximum in the close vicinity of the pixel-level cross-correlation peak.

### 4.1. Example 1

The first counterexample with synthetic images was chosen for its simplicity so that it can be easily reproduced. The Matlab code for generating the presented result is available for download [16]. To show the robustness of this counterexample, the Matlab code can be optionally run with some random noises added to the generated synthetic images, though the result presented below is noise-free. Counterexamples based on true images (as presented in Examples 2 and 3) are also included in [16].

Consider two binary images of 18×18 pixels as shown in Figure 3, with the template (ROI) chosen as the red square window of 8×8 pixels in each image. The template in the first image contains a diagonal pattern. In the second image, this diagonal pattern is shifted by one pixel toward the right and also by one pixel toward the bottom. In these binary images, the intensity is 1 at the darker pixels and 10 at the brighter pixels.

The normalized 3×3 cross-correlation matrix around the peak as defined in (Equation 8) is:(12)$$\begin{array}{c} \Gamma =\left[\begin{array}{ccc} 0.2236 & \;0.2236\; & 0.8059\\ 0.2236 & \;1\; & 0.2236\\ 0.8059 & \;0.2236\; & 0.2236 \end{array}\right]. \end{array}$$

Fitting the second degree polynomial $p_{\theta }\left(x,y\right)$ to Γ for (x,y)∈G by solving the least squares problem (Equation 11) yields the solution:(13)$$\begin{array}{c} \theta^{*}=\left[\begin{array}{c} 0.5256\\ 0.0000\\ 0.0000\\ -0.0647\\ 0.2911\\ -0.0647 \end{array}\right]. \end{array}$$

The corresponding quadratic surface $p_{\theta^{*}}\left(x,y\right)$ exhibits a saddle point, as illustrated in Figure 4. It has no global maximum, despite the fact that the central entry of the matrix Γ is its largest entry.

This counterexample clearly invalidates the widespread intuition that the fitted polynomial $p_{\theta }\left(x,y\right)$ in the QSF method always has a global maximum.

### 4.2. Example 2

This example is based on true images of the moon, included in Matlab as a part of its image examples. The Matlab code generating the presented results are available for download [16]. Two images of the moon with 537×358 pixels are shown in Figure 5. As in Example 1, with some chosen template (ROI), cross-correlations between the two images are computed, and a quadratic surface is fitted to the normalized 3×3 cross-correlation matrix around the peak. In most situations, the fitted surface has a maximum close to the pixel-level cross-correlation peak, but unexpected cases do happen with some particular template choices.

For the chosen template illustrated by the red windows in Figure 5, the fitted quadratic surface is shown in Figure 6. The surface exhibiting a saddle point has no maximum. This result can be reproduced with the Matlab code downloadable from [16].

### 4.3. Example 3

This example is based the same images of the moon as in Example 2, but with another choice of the template for cross-correlation computation, as illustrated by the red windows in Figure 7. The fitted quadratic surface shown in Figure 8 has a maximum located at
$$\left(x^{*},y^{*}\right)=\left(13.76,4.28\right),$$
as indicated by the cyan dot on the surface. This maximum is far outside the square area of the one-pixel vicinity of the pixel-level cross-correlation peak delimited by the red dotted lines on the bottom plane in Figure 8. As a subpixel refinement, the maximum of the fitted surface should satisfy $|x^{*}|<1$ and $|y^{*}|<1$, which is not the case in this example. The details of this result are available from [16].

The examples presented in this section confirm that, in some rare situations, the quadratic surface fitted in the QSF method does not have a maximum, or has a maximum far away from the square area of the one-pixel vicinity of the pixel-level cross-correlation peak.

After the answer to the first question raised at the end of Section 2, the following sections will then answer the other two remaining questions.

## 5. Conditions for the Existence of a Maximum

In elementary algebra, it is well known that the second degree polynomial $p_{\theta }\left(x,y\right)$ as expressed in (Equation 10) has a unique global maximum if its Hessian matrix,
(14)$$\begin{array}{c} H≜\left[\begin{array}{cc} \theta_{4} & \theta_{5}/2\\ \theta_{5}/2 & \theta_{6}, \end{array}\right] \end{array}$$
is negative definite [17]. However, this simply stated fact does not directly help us to understand how the normalized cross-correlation values γ(n,m) (those filling up the Γ matrix in (Equation 8)) should be, so that the fitted $p_{\theta }\left(x,y\right)$ has a global maximum. Because the polynomial coefficients $\theta_{1},\cdots ,\theta_{6}$ are determined from the values of γ(n,m) by solving the least squares problem (Equation 11), it is straightforward to express the negative definiteness condition of *H* in terms of γ(n,m). Then, in principle, the condition for the existence of a global maximum of $p_{\theta }\left(x,y\right)$ will be formulated in terms of the normalized cross-correlation values γ(n,m). Nevertheless, this approach results in sophisticated conditions, notably an inequality involving the determinant of *H* expressed in terms of γ(n,m). For a better understanding, the result presented below will be formulated with simple and easily interpretable inequalities about the values γ(n,m) filling up Γ. For instance, one of these simple inequalities states that the central entry γ(0,0) of Γ is its largest entry. As shown by the previously presented counterexamples, this condition alone is not sufficient. It is then completed by similar simple inequalities. 

**Theorem** **1.**
*If the normalized cross-correlation values γ(n,m) filling up the matrix *Γ* satisfy:*

(15)
$$\begin{array}{cc} \gamma (0,0) & \geq \gamma (n,m)\;\;for\;all\;n,m\in \{-1,0,+1\} \end{array}$$


(16)
$$\begin{array}{cc} \gamma (0,m) & >\gamma (n,m)\;\;for\;all\;n,m\in \{-1,+1\} \end{array}$$


(17)
$$\begin{array}{cc} \gamma (n,0) & >\gamma (n,m)\;\;for\;all\;n,m\in \{-1,+1\}, \end{array}$$

*then the second degree polynomial $p_{\theta }\left(x,y\right)$ fitted to the entries of *Γ* by solving the least squares problem *(Equation 11)* has a unique global maximum.*


Interpretation of the conditions of Theorem 1. 

Inequalities (Equation 15): the central entry γ(0,0) has the largest value among all the 9 entries of Γ.Inequalities (Equation 16): the middle entry γ(0,±1) is the largest entry of the top or the bottom row of Γ.Inequalities (Equation 17): the middle entry γ(±1,0) is the largest entry of the right or the left column of Γ. 

**Proof** **of** **Theorem** **1.**In order to shorten lengthy equations and inequalities, let us introduce more compact notations for the normalized cross-correlation values γ(m,m) filling up the matrix Γ defined in (Equation 8), so that Γ is rewritten as:
(18)$$\begin{array}{c} \Gamma =\left[\begin{array}{ccc} a & \;e\; & b\\ f & \;i\; & g\\ c & \;h\; & d \end{array}\right]. \end{array}$$Remark that the letters a,b,d,⋯,i fill Γ first at the four corners, then the middles of side rows and columns, before finishing at the central entry.With these compact notations, the least squares solution (Equation 11) leads to
(19a)$$\begin{array}{cc} 9\theta_{1} & =2(e+f+g+h)+5i-(a+b+c+d) \end{array}$$
(19b)$$\begin{array}{cc} 6\theta_{2} & =(b-a)+(g-f)+(d-c) \end{array}$$
(19c)$$\begin{array}{cc} 6\theta_{3} & =(a-c)+(e-h)+(b-d)\\ -6\theta_{4} & =(e-a)+(e-b)+(h-c)+(h-d) \end{array}$$
(19d)$$\begin{array}{c} \;\;\;\;+(i-f)+(i-g) \end{array}$$
(19e)$$\begin{array}{cc} 4\theta_{5} & =(b-a+c-d)\\ -6\theta_{6} & =(f-a)+(f-c)+(g-b)+(g-d) \end{array}$$
(19f)$$\begin{array}{c} \;\;\;\;+(i-e)+(i-h). \end{array}$$As already mentioned in this paper, the negative definiteness of the Hessian matrix *H* defined in (Equation 14) ensures that the polynomial $p_{\theta }\left(x,y\right)$ has a global maximum. Based on Sylvester’s criterion (Usually Sylvester’s criterion [18] is about the positive definiteness of a real symmetric (or complex Hermitian) matrix. It is trivial to translate this criterion to the case of negative definiteness; this negative definiteness will be checked through:
(20)$$\begin{array}{cc} \theta_{4} & <0 \end{array}$$
(21)$$\begin{array}{cc} \det(H) & >0. \end{array}$$According to the inequalities assumed in (Equation 16), *e* (or *h*, resp.) is the largest entry of the top (or bottom, resp.) row of Γ, then
(22)$$\begin{array}{cc} \; & e-a>0,\;\;\;e-b>0 \end{array}$$
(23)$$\begin{array}{cc} \; & h-c>0,\;\;\;h-d>0 \end{array}$$
and according to (Equation 15), *i* is the largest entry of Γ, then
(24)$$\begin{array}{c} i-f\geq 0,\;\;\;i-g\geq 0. \end{array}$$These inequalities together with ([Disp-formula FD19d-sensors-22-01274]) immediately imply (Equation 20).It is more involved to check (Equation 21). The inequalities in (Equation 22) imply:
(25)$$\begin{array}{c} \left(1+\frac{3}{4}\right)\left(e-a\right)>0>\left(-1+\frac{3}{4}\right)\left(e-b\right), \end{array}$$
then
(26)$$\begin{array}{cc} \left(e-a)+(e-b\right) & >-\frac{3}{4}\left(e-a\right)+\frac{3}{4}\left(e-b\right) \end{array}$$
(27)$$\begin{array}{cc} \; & =-\frac{3}{4}\left(b-a\right). \end{array}$$Repeat the reasoning from (Equation 25) to (Equation 27) while interchanging the positions of (e−a) and (e−b):
(28)$$\begin{array}{c} \left(1+\frac{3}{4}\right)\left(e-b\right)>0>\left(-1+\frac{3}{4}\right)\left(e-a\right), \end{array}$$
leading to
(29)$$\begin{array}{cc} \left(e-a)+(e-b\right) & >\frac{3}{4}\left(b-a\right). \end{array}$$Combining (Equation 27) and (Equation 29) yields
(30)$$\begin{array}{cc} \left(e-a)+(e-b\right) & >\frac{3}{4}\left|b-a\right|. \end{array}$$This result expresses a relationship between the entries in the top row of the matrix Γ. A similar reasoning then leads to the following relationship between the entries in the bottom row of Γ:
(31)$$\begin{array}{cc} \left(h-c)+(h-d\right) & >\frac{3}{4}\left|c-d\right|. \end{array}$$According to the inequalities assumed in (Equation 15), *i* is the largest entry of Γ, then
(32)$$\begin{array}{cc} \; & i-f\geq 0,\;\;\;i-g\geq 0. \end{array}$$Take the sums of the respective sides of the four inequalities in (Equation 30)–(Equation 32), then
$$\begin{array}{cc} \; & \left(e-a)+(e-b)+(h-c)+(h-d)+(i-f)+(i-g\right) \end{array}$$
(33)$$\begin{array}{cc} & \;\;\;\;\;\;>\frac{3}{4}|b-a|+\frac{3}{4}\left|c-d\right| \end{array}$$
(34)$$\begin{array}{cc} \; & \;\;\;\;\;\;\geq \frac{3}{4}\left|b-a+c-d\right|. \end{array}$$This result then implies that $-6\theta_{4}$ and $4\theta_{5}$, as expressed respectively in ([Disp-formula FD19d-sensors-22-01274]) and ([Disp-formula FD19e-sensors-22-01274]), satisfy:
(35)$$\begin{array}{c} -6\theta_{4}>\frac{3}{4}\left|4\theta_{5}\right|, \end{array}$$
hence
(36)$$\begin{array}{c} -\theta_{4}>\frac{1}{2}\left|\theta_{5}\right|\geq 0. \end{array}$$Following the same approach, it is then similarly shown that
(37)$$\begin{array}{c} -\theta_{6}>\frac{1}{2}\left|\theta_{5}\right|\geq 0. \end{array}$$This last result can also be deduced from a certain “symmetry” between the formulae expressing $\theta_{4}$ and $\theta_{6}$ in ([Disp-formula FD19d-sensors-22-01274]) and ([Disp-formula FD19f-sensors-22-01274]).It then follows from (Equation 14), (Equation 36) and (Equation 37) that
(38)$$\begin{array}{c} \det\left(H\right)=\theta_{4}\theta_{6}-\frac{1}{4}\theta_{5}^{2}>0. \end{array}$$Therefore, the two inequalities (Equation 20) and (Equation 21) ensuring the negative definiteness of *H* are successfully checked. It is then established that the second degree polynomial $p_{\theta }\left(x,y\right)$ has a unique global maximum. □

When the fitted polynomial $p_{\theta }\left(x,y\right)$ has a unique global maximum, it may happen that this maximum is far away from the origin (x,y)=(0,0) corresponding to the optimized integer shifts $\left({\tilde{n}}^{*},{\tilde{m}}^{*}\right)$, outside the square area of the one-pixel vicinity of the cross-correlation peak. Such situations are not desirable, since the subpixel refinement should not modify the estimated shifts by more than one pixel. The following result ensures that the maximum of $p_{\theta }\left(x,y\right)$ stays inside the one-pixel vicinity, under easily interpretable conditions.

**Theorem** **2.**
*If, in addition to the conditions of Theorem 1, the normalized cross-correlation values γ(n,m) filling up the matrix *Γ* satisfy, for all n,m∈{−1,+1},*

(39)
$$\begin{array}{cc} \gamma (0,m)-\gamma (n,m) & >\frac{1}{5}\left[\gamma (0,m)-\gamma (-n,m)\right] \end{array}$$


(40)
$$\begin{array}{cc} \gamma (n,0)-\gamma (n,m) & >\frac{1}{5}\left[\gamma (n,0)-\gamma (n,-m)\right], \end{array}$$

*then the maximum of the fitted polynomial $p_{\theta }\left(x,y\right)$ is located at $\left(x^{*},y^{*}\right)$ such that $|x^{*}|<1$ and $|y^{*}|<1$.*


Interpretation of the conditions of Theorem 2. 

The conditions inherited from Theorem 1 ensure that the middle entry in each row or column of Γ is the largest entry of the row or column, without imposing any “degree of symmetry”. For example, among the top row of Γ as expressed in (Equation 8), inequalities formulated in (Equation 16) ensure that γ(0,1) is the largest entry, but the ratio [γ(0,1)−γ(−1,1)/[γ(0,1)−γ(1,1)] can be any positive number. Two of the inequalities in the extra condition (Equation 39) of Theorem 2 constrain this ratio between 1/5 and 5, thus limiting the dissymmetry between γ(−1,1) and γ(1,1). 

**Proof** **of** **Theorem** **2.**Based on Theorem 1 (its conditions are inherited here), the fitted polynomial $p_{\theta }\left(x,y\right)$ has a unique global maximum in $R^{2}$, which is located at
(41)$$\begin{array}{c} x^{*}=\frac{2\theta_{2}\theta_{6}-\theta_{3}\theta_{5}}{\theta_{5}^{2}-4\theta_{4}\theta_{6}} \end{array}$$
(42)$$\begin{array}{c} y^{*}=\frac{2\theta_{3}\theta_{4}-\theta_{2}\theta_{5}}{\theta_{5}^{2}-4\theta_{4}\theta_{6}}. \end{array}$$In order to prove $|x^{*}|<1$, it will be shown that the numerator of $|x^{*}|$ is smaller than its denominator. The proof for proving $|y^{*}|<1$ will be made similarly.With the compact notations filling up the matrix Γ introduced in (Equation 18) for the normalized correlation values γ(n,m), one of the inequalities contained in (Equation 39), namely
(43)$$\begin{array}{cc} \gamma (0,1)-\gamma (-1,1) & >\frac{1}{5}\left[\gamma (0,1)-\gamma (1,1)\right], \end{array}$$
is translated into
(44)$$\begin{array}{c} e-a>\frac{1}{5}\left(e-b\right), \end{array}$$
which is rewritten as
(45)$$\begin{array}{c} \frac{5}{2}\left(e-a\right)>\frac{1}{2}\left(e-b\right), \end{array}$$
or in a slightly different form
(46)$$\begin{array}{c} \left(1+\frac{3}{2}\right)\left(e-a\right)>\left(-1+\frac{3}{2}\right)\left(e-b\right). \end{array}$$Then
(47)$$\begin{array}{cc} \left(e-a)+(e-b\right) & >\frac{3}{2}\left(e-b\right)-\frac{3}{2}\left(e-a\right) \end{array}$$
(48)$$\begin{array}{cc} \; & =\frac{3}{2}\left(a-b\right). \end{array}$$A similar reasoning (by interchanging the positions of (e−a) and (e−b)) leads to
(49)$$\begin{array}{cc} \left(e-b)+(e-a\right) & >\frac{3}{2}\left(e-a\right)-\frac{3}{2}\left(e-b\right) \end{array}$$
(50)$$\begin{array}{cc} \; & =-\frac{3}{2}\left(a-b\right). \end{array}$$Combining (Equation 48) and (Equation 50) then amounts to
(51)$$\begin{array}{c} \left(e-a\right)+\left(e-b\right)>\frac{3}{2}\left|b-a\right|. \end{array}$$This last inequality concerns the entries in the top row of Γ as in (Equation 18). Similar reasonings about the bottom row, the left and right columns of Γ then lead to
(52)$$\begin{array}{cc} \left(h-c)+(h-d\right) & >\frac{3}{2}\left|c-d\right|. \end{array}$$
(53)$$\begin{array}{cc} \left(f-a)+(f-c\right) & >\frac{3}{2}\left|a-c\right|. \end{array}$$
(54)$$\begin{array}{cc} \left(g-b)+(g-d\right) & >\frac{3}{2}\left|b-d\right|.. \end{array}$$Because *i* is the largest entry of Γ (condition inherited from Theorem 1),
(55)$$\begin{array}{c} i-f\geq 0,\;\;\;i-d\geq 0. \end{array}$$Summarizing (Equation 51), (Equation 52) and (Equation 55) then implies that $\theta_{4}$ as expressed in ([Disp-formula FD19d-sensors-22-01274]) satisfies:
$$\begin{array}{cc} -6\theta_{4} & =(e-a)+(e-b)+(h-c)+(h-d) \end{array}$$
(56)$$\begin{array}{cc} & \;\;\;\;+(i-f)+(i-g) \end{array}$$
(57)$$\begin{array}{cc} \; & >\frac{3}{2}|b-a|+\frac{3}{2}\left|c-d\right| \end{array}$$
(58)$$\begin{array}{cc} \; & \geq \frac{3}{2}\left|b-a+c-d\right|. \end{array}$$This result, together with ([Disp-formula FD19e-sensors-22-01274]), then yields to:
(59)$$\begin{array}{c} -\theta_{4}>\left|\theta_{5}\right|\geq 0. \end{array}$$Some similar reasonings (due to a certain “symmetry” between $\theta_{4}$ and $\theta_{6}$ in (19) then lead to
(60)$$\begin{array}{c} -\theta_{6}>\left|\theta_{5}\right|\geq 0. \end{array}$$Add together the respective sides of ([Disp-formula FD19b-sensors-22-01274]) and ([Disp-formula FD19d-sensors-22-01274]), then,
(61)$$\begin{array}{c} 6\left(\theta_{2}-\theta_{4}\right)=2\left[(e-a)+(h-c)+(i-f)\right]. \end{array}$$Every parenthesis at the right hand side of (Equation 61) is positive, due to conditions inherited from Theorem 1. Then,
(62)$$\begin{array}{c} \theta_{2}-\theta_{4}>0. \end{array}$$Similarly,
(63)$$\begin{array}{cc} 6(\theta_{2}+\theta_{4}) & =-2\left[(e-b)+(h-d)+(i-g)\right]<0. \end{array}$$Based on the last 2 inequalities, $-\theta_{4}$ satisfies:
(64)$$\begin{array}{c} -\theta_{4}>\pm \theta_{2}, \end{array}$$
hence
(65)$$\begin{array}{c} -\theta_{4}>\left|\theta_{2}\right|\geq 0. \end{array}$$In the same manner, combining ([Disp-formula FD19c-sensors-22-01274]) and ([Disp-formula FD19f-sensors-22-01274]) yields:
(66)$$\begin{array}{c} 6\left(\theta_{3}-\theta_{6}\right)=2\left[(f-c)+(g-d)+(i-h)\right]>0 \end{array}$$
(67)$$\begin{array}{c} 6\left(\theta_{3}+\theta_{6}\right)=-2\left[(f-a)+(g-b)+(i-e)\right]<0, \end{array}$$
then
(68)$$\begin{array}{c} -\theta_{6}>\left|\theta_{3}\right|\geq 0. \end{array}$$The following steps of the proof will be essentially based on:
(69)$$\begin{array}{cc} |\theta_{4}| & >|\theta_{5}| \end{array}$$
(70)$$\begin{array}{cc} |\theta_{6}| & >|\theta_{5}| \end{array}$$
(71)$$\begin{array}{cc} |\theta_{4}| & >|\theta_{2}| \end{array}$$
(72)$$\begin{array}{cc} |\theta_{6}| & >|\theta_{3}|, \end{array}$$
respectively, due to (Equation 59), (Equation 60), (Equation 65) and (Equation 68).On the one hand, (Equation 71), (Equation 72) and (Equation 69) imply (the step leading to (Equation 74) below)
(73)$$\begin{array}{cc} |2\theta_{2}\theta_{6}-\theta_{3}\theta_{5}|\; & \leq 2|\theta_{2}\left|\right|\theta_{6}\left|+\right|\theta_{3}\left|\right|\theta_{5}| \end{array}$$
(74)$$\begin{array}{cc} \; & <2|\theta_{4}\left|\right|\theta_{6}\left|+\right|\theta_{6}\left|\right|\theta_{4}| \end{array}$$
(75)$$\begin{array}{cc} \; & =3|\theta_{4}\left|\right|\theta_{6}|; \end{array}$$
and on the other hand, (Equation 69) and (Equation 70) lead to (the step leading to (Equation 78) below)
(76)$$\begin{array}{cc} |\theta_{5}^{2}-4\theta_{4}\theta_{6}| & \geq |4\theta_{4}\theta_{6}\left|-\right|\theta_{5}^{2}| \end{array}$$
(77)$$\begin{array}{cc} \; & =4|\theta_{4}\left|\right|\theta_{6}\left|-\right|\theta_{5}\left|\right|\theta_{5}| \end{array}$$
(78)$$\begin{array}{cc} \; & >4|\theta_{4}\left|\right|\theta_{6}\left|-\right|\theta_{4}\left|\right|\theta_{6}| \end{array}$$
(79)$$\begin{array}{cc} \; & =3|\theta_{4}\left|\right|\theta_{6}|. \end{array}$$Therefore
(80)$$\begin{array}{c} |2\theta_{2}\theta_{6}-\theta_{3}\theta_{5}\left|<\right|\theta_{5}^{2}-4\theta_{4}\theta_{6}|. \end{array}$$It is then concluded that $x^{*}$ as expressed in (Equation 41) satisfies
(81)$$\begin{array}{c} |x^{*}|<1. \end{array}$$In the same way, it is also proved that:
(82)$$\begin{array}{c} |y^{*}|<1. \end{array}$$The proof of Theorem 2 is thus completed. □

## 6. Handling Failures of the QSF Method

There are two possible failure cases:Case 1, the fitted polynomial $p_{\theta }\left(x,y\right)$ has no global maximum;Case 2, $p_{\theta }\left(x,y\right)$ has a global maximum, reached at $\left(x^{*},y^{*}\right)$, but $|x^{*}|\geq 1$ and/or $|y^{*}|\geq 1$.

In the first case, the only reasonable proposition is to retain the optimized integer shifts $\left({\tilde{n}}^{*},{\tilde{m}}^{*}\right)$ as the estimated total shifts.

In the second case, solutions on the boundary satisfying $|x^{*}|\leq 1$ and $|y^{*}|\leq 1$ are accepted. If $|x^{*}|>1$ and/or $|y^{*}|>1$, then the subpixel shifts are estimated by solving the constrained optimization problem:(83)$$\begin{array}{c} \left(x^{★},y^{★}\right)=\arg\max_{|x|\leq 1,|y|\leq 1}p_{\theta }\left(x,y\right), \end{array}$$
and the estimated total shifts amount to:(84)$$\begin{array}{c} ({\tilde{n}}^{*}+x^{★},{\tilde{m}}^{*}+y^{★}). \end{array}$$

As $p_{\theta }\left(x,y\right)$ is a second degree polynomial, the constrained optimization problem (Equation 83) can be solved by quadratic programming algorithms. In this considered Case 2, the unique unconstrained maximum of $p_{\theta }\left(x,y\right)$ is outside the square area constrained by |x|≤1 and |y|≤1, hence the constrained solution is certainly on the boundary of the constrained square area, on one of its sides or on one of its corners. Given the simplicity of this quadratic problem, instead of applying a general quadratic programming tool, the constrained optimization problem (Equation 83) can be solved as follows:Compute the values of $p_{\theta }\left(x,y\right)$ at the four corners of the square, namely $p_{\theta }\left(\pm 1,\pm 1\right)$;Find the maximums of $p_{\theta }\left(x,\pm 1\right)$ in *x* and $p_{\theta }\left(\pm 1,y\right)$ in *y*:
(85)$$\begin{array}{c} x_{\pm }^{★}=\arg\max_{x\in R}p_{\theta }\left(x,\pm 1\right)=-\frac{\theta_{2}\pm \theta_{5}}{2\theta_{4}}; \end{array}$$
(86)$$\begin{array}{c} y_{\pm }^{★}=\arg\max_{y\in R}p_{\theta }\left(\pm 1,y\right)=-\frac{\theta_{3}\pm \theta_{5}}{2\theta_{6}}. \end{array}$$Eliminate any result(s) not satisfying $|x_{\pm }^{★}|\leq 1$ or $|y_{\pm }^{★}|\leq 1$.Find the maximum value among the four corner values $p_{\theta }\left(\pm 1,\pm 1\right)$ and the non-eliminated values $p_{\theta }\left(x_{\pm }^{★},\pm 1\right)$ and/or $p_{\theta }\left(\pm 1,y_{\pm }^{★}\right)$, if any. The solution of the constrained optimization problem (Equation 83) is then given by the location of this maximum.

## 7. Assessment Based on Two Typical Types of Images

The original QSF method and the modified method are tested with two typical types of images for failure rate evaluation. As already stated in the introduction, the original QSF method works correctly in most situations. The occurrence frequency of its failures depends on the type of processed images. Two typical examples are presented below in this section.

For the moon example already considered in Section 4, when the template window is close to a corner of the left side of the images, as shown in Figure 5, the selected region of interest has a diagonally dominant pattern, sharing a common characteristic with the synthetic example shown in Figure 3. For this reason, the QSF method is more likely to encounter difficulties with these images.

A second example comes from the publicly available DIC Challenge database [19,20] provided by the Society for Experimental Mechanics. The tested images contain uniform patterns, as shown in Figure 9.

For each trial with the original QSF method, the template window is placed at a different position in the processed images, as illustrated in Figure 5 and Figure 9. Among all the trials, the number of cases where the fitted quadratic surface exhibits a saddle point (absence of maximum) is counted. The second type of failures, with maximum outside the one-pixel vicinity, is also counted. The results are summarized in Table 1.

As expected, the moon example leads to more failures (0.21% and 4.79% of the two types of failures over the total number of trials) due to the existence of diagonally dominant patterns. For the second type of failures, the maximum of the fitted surface can be far away from the the one-pixel vicinity, up to 52 pixels. On the other hand, the DIC Challenge example with uniform patterns has very few failures (8.74 ppm and 52.4 ppm).

Though failures of the original QSF method rarely happen, it is important to prevent them for reliable applications.

After modifications of the QSF method as proposed in Section 6, for each the trials with the above two examples, the modified method successfully finds a subpixel displacement $\left(x^{*},y^{*}\right)$ such that $|x^{*}|<1$ and $|y^{*}|<1$.

## 8. Conclusions

In this paper, within the scope of digital image correlation techniques, some theoretical aspects of the QSF method have been investigated. It has been shown that, contrary to a widespread intuition, the quadratic surface fitted in the QSF method does not always have a maximum. Then, for a better understanding of this method, it is analyzed by providing mathematical conditions ensuring expected results. Algorithm modifications have also been proposed to handle unexpected cases. Finally, experimental results based on two typical types of images have been reported. These results will contribute to consolidating both the theory and the practice of the QSF method.

## Figures and Tables

**Figure 1 sensors-22-01274-f001:**
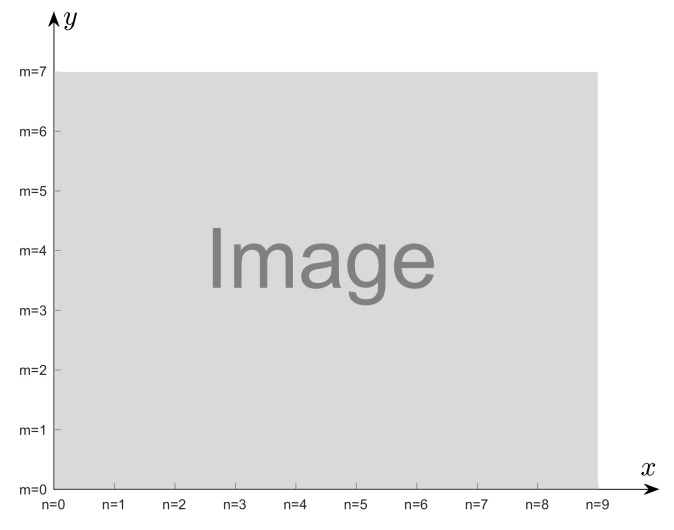
Illustration of the Cartesian coordinate system and integer coordinates (n,m).

**Figure 2 sensors-22-01274-f002:**
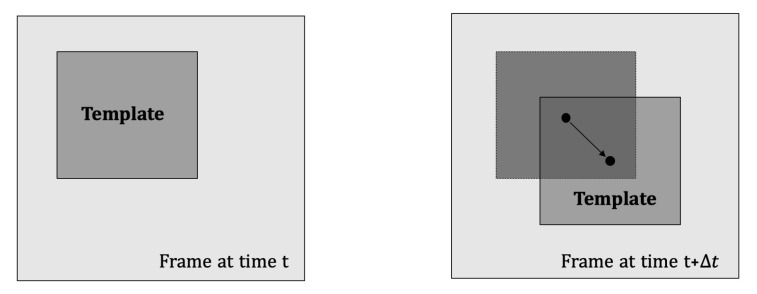
Template shifts between two successive images.

**Figure 3 sensors-22-01274-f003:**
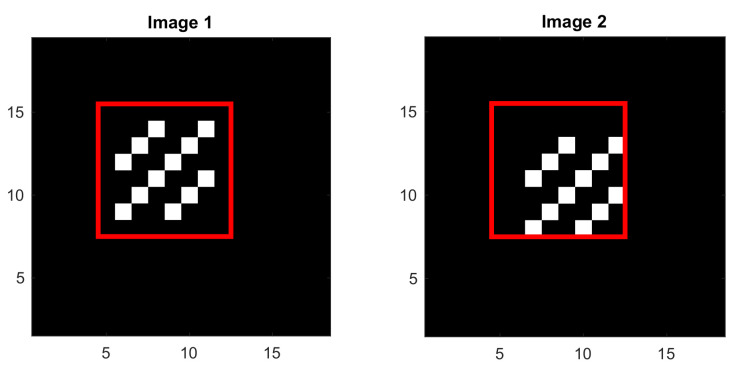
Two synthetic images of 18×18 pixels. The intensity is 1 at the darker pixels and 10 at the brighter pixels, The template (ROI) is selected as the red square window of 8×8 pixels.

**Figure 4 sensors-22-01274-f004:**
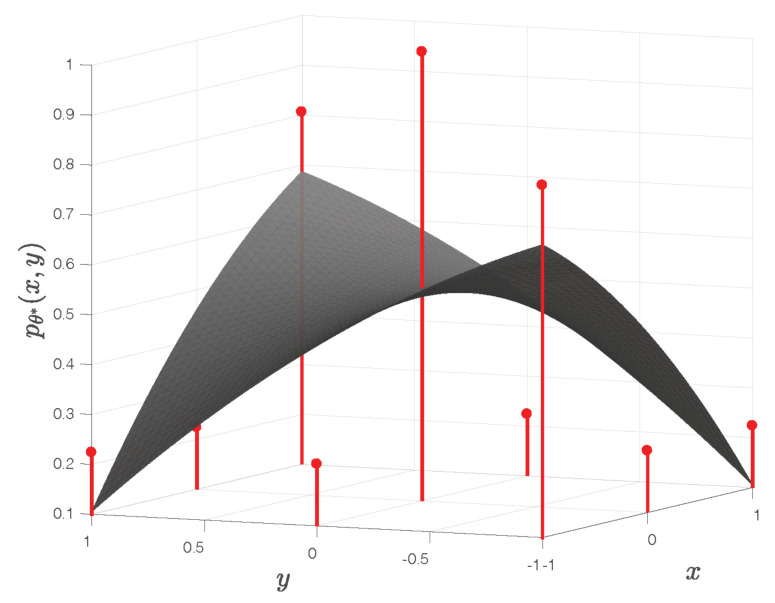
For the images in Figure 3, the fitted quadratic surface exhibits a saddle point. The vertical red line segments represent the entries of the normalized cross-correlation matrix Γ.

**Figure 5 sensors-22-01274-f005:**
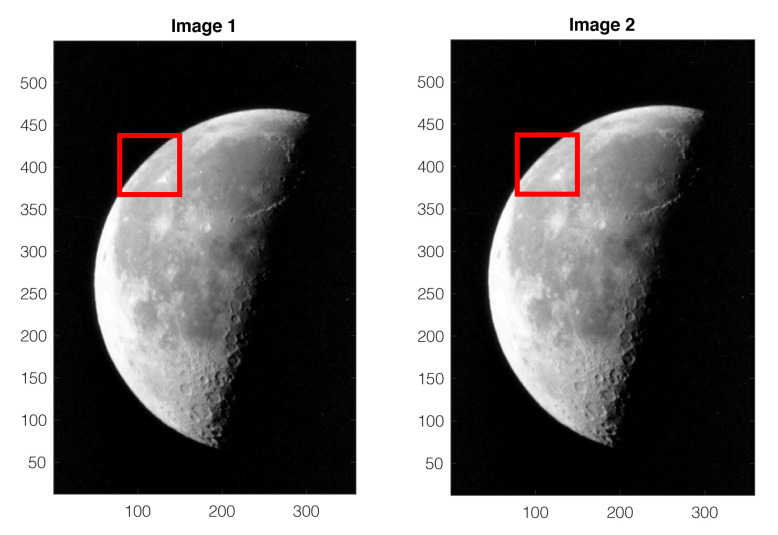
Two images of the moon of 537×358 pixels. The template is chosen as the red windows.

**Figure 6 sensors-22-01274-f006:**
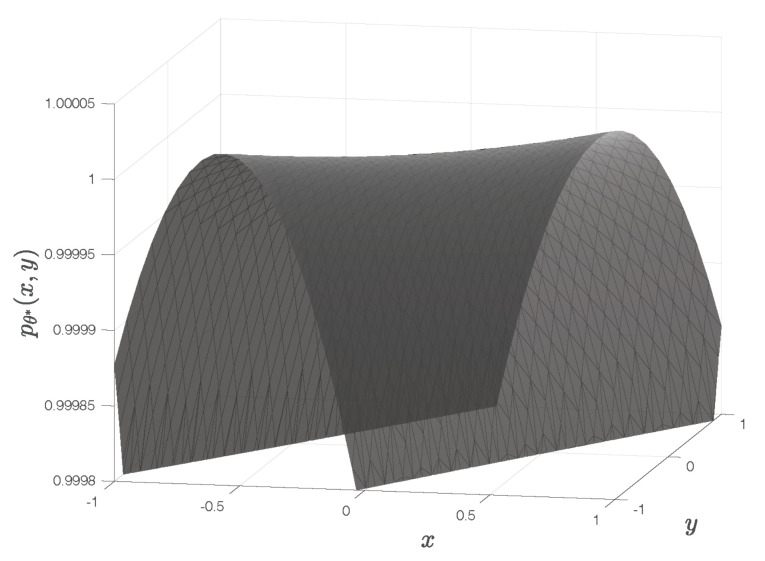
For the images and template in Figure 5, the fitted quadratic surface exhibits a saddle point.

**Figure 7 sensors-22-01274-f007:**
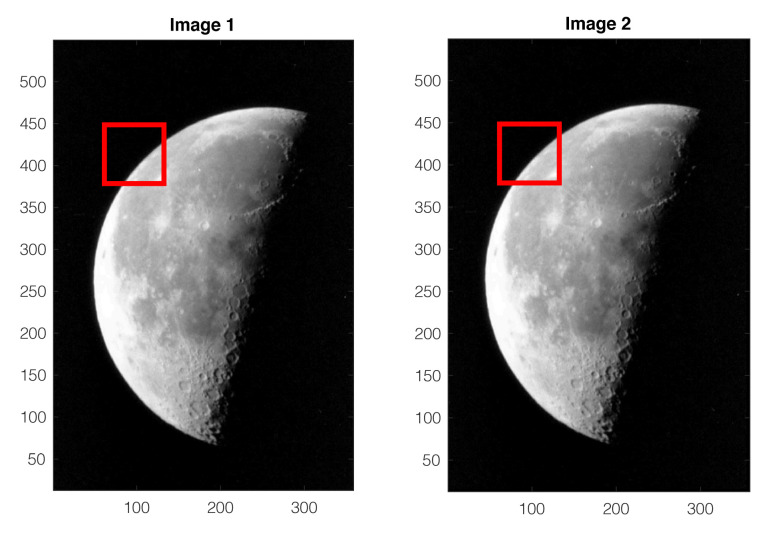
Two images of the moon, the same as in Figure 5, but with a different template choice shown by the red windows.

**Figure 8 sensors-22-01274-f008:**
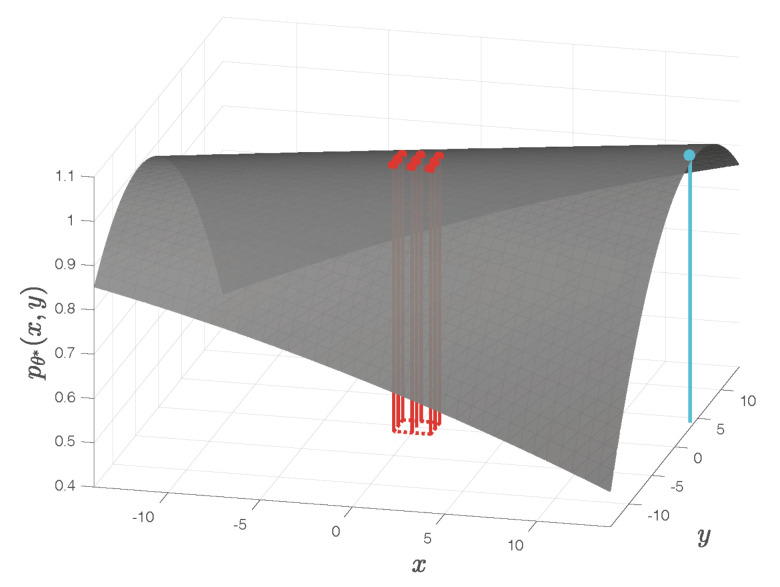
For the images and template in Figure 7, the fitted quadratic surface has a maximum located at $\left(x^{*},y^{*}\right)=\left(13.76,4.28\right)$, as shown by the cyan dot on the surface. The red dotted lines on the bottom plane delimit the square area of the one-pixel vicinity around the pixel-level cross-correlation peak. As a subpixel refinement, the maximum of the fitted surface should satisfy $|x^{*}|<1$ and $|y^{*}|<1$, which is not the case in this example. See [16] for the details of this example.

**Figure 9 sensors-22-01274-f009:**
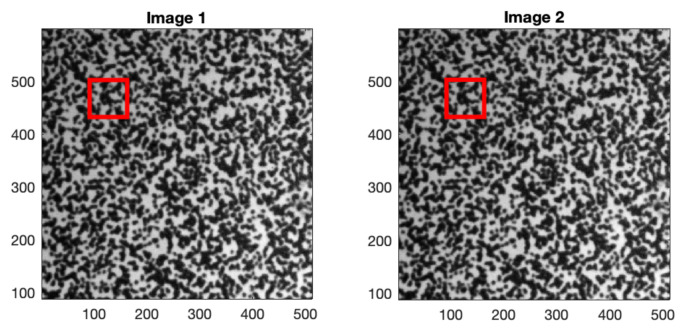
Two images of the DIC Challenge database [19], each with resolution 512 × 512 pixels. The the red windows correspond to an example of template for the QSF method.

**Table 1 sensors-22-01274-t001:** Original QSF method failure occurrences in the Moon and DIC Challenge examples. Total trials: number of total trials of the QSF method. Absence of maximum: number of trials yielding a saddle point (and in percentage of total trials). Maximum outside the one-pixel vicinity: number of trials yielding a maximum of the fitted surface located at $\left(x^{*},y^{*}\right)$ with $|x^{*}|>1$ or $|y^{*}|>1$ (and ppm, part per million). $\max|x^{*}|$: the maximum value of $|x^{*}|$ among all the trials. $\max|y^{*}|$: the maximum value of $|y^{*}|$ among all the trials.

	Total Trials	Absence of Maximum	Maximum Outside One-Pixel Vicinity	$\max|x^{*}|$	$\max|y^{*}|$
Moon	10,938	23 (0.21%)	524 (4.79%)	35.9474	52.0165
DIC Challenge	114,444	1 (8.74 ppm)	6 (52.4 ppm)	4.7301	4.7301

## Data Availability

Not applicable.

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
