# Peer review of "On Quadratic Interpolation of Image Cross-Correlation for Subpixel Motion Extraction†"

_sensors, 2022, doi:10.3390/s22031274_

Round 1

Reviewer 1 Report

This manuscript analyzed a Quadratic Surface Fitting (QSF) method for subpixel motion extraction mathematically so it found two failure cases of the QSF method and presented an improved QSF algorithm to overcome these failure cases. However, results of applying the improved algorithm compared to the original algorithm were not presented.

1. The subpixel motion extraction algorithm through the proposed QSF method needs a comparison with the original QSF algorithm.
2. A summary of the improvement of the QSF algorithm should be added in the Introduction section.
3. Line 58-61 in the Introduction section refers to a previous conference paper, but that seems to be unnecessary.
4. Italics are being abused in the manuscript. Only use italics for words that require italics, e.g., variables.
5. There are some French words, e.g., négligeable. Please correct these to English words.

Author Response

Please see our answers in the uploaded PDF file.

Reviewer 2 Report

TYPOS
=====

can be reproduced with the mages and Matlab => can be reproduced with the images and Matlab
faraway => far away
non eliminated values => non-eliminated values

CONTENT
=======

The authors use both $\rho$ and $p$ in their mathematical notation, which is not incorrect, but which may be somewhat problematic if a reader is, e.g., short-sighted, so maybe these symbols should be somewhat changed due to their current relatively high similarity. This is only a friendly suggestion and it is definitely not mandatory.

The content of the paper is generally well presented and it touches an interesting topic. While the topic is not necessarily critical and it represents a special case, it is good that it is finally covered in a proper and sound way. Since it may also be highly useful to practitioners, I would suggest to the authors to also empirically calculate how often are on average the conditions in the two theorems met in real-world images. For example, the authors could calculate this on any publicly available dataset just to give the readers a feeling on how often the special case that is in the focus of this paper appears. This would definitely significantly improve the practical value of the paper, whose theoretical value is already firmly established. Namely, in my opinion the practitioners would be able to better decisions based on well prepared information if these calculations were carried out and presented in the paper here.

OVERALL
=======

In my opinion the paper should be accepted after some minor changes have been made due to its practical and theoretical usefulness.

Author Response

(The authors gave the same response as above.)

Round 2

Reviewer 1 Report

It was confirmed that the manuscript was revised according to the comments. However, some modifications are needed as follows.

  1. In Section 7, the division into subsections seems unnecessary. The content of Section 7.1 is similar to Section 4 'Counterexamples'. Describe the sub-pixel estimation in the motion extraction by the proposed method compared with the failure case of the original QSF method. 
  2. Most of the unnecessary italics have been corrected to normal-style. However, some adverb clauses are still italicized, e.g., page 1 line 26 ‘e.g.’, page 4 line 102 ‘implicitly’, and line 108, ‘always’. please correct these to normal-style.
  3. In Introduction section, a description of Section 7 needs to be added to the last paragraph.

Author Response

Please see the uploaded PDF file.
